# Experimental Study and Analysis on Workability and Mechanical Performance of High Fluidity Recycled Concrete

**DOI:** 10.3390/ma15176104

**Published:** 2022-09-02

**Authors:** Cun Hui, Yan Liu, Ran Hai, Mingliang Liu

**Affiliations:** 1School of Architecture and Civil Engineering, Zhongyuan University of Technology, Zhengzhou 450007, China; 2Shaanxi Architecture Science Research Institute Co., Ltd., Xi’an 710082, China

**Keywords:** recycled concrete, recycled aggregate, sand ratio, workability, mechanical properties

## Abstract

In order to study the workability and mechanical performance of high fluidity recycled concrete, parallel tests were carried out to prepare high fluidity recycled concrete by changing the amount of recycled aggregate (20%, 30%, 40%, 50% and 60%) and the sand ratio (0.37, 0.38, 0.39 and 0.40). The fluidity, compression strength, tensile strength and flexural strength of each mix were tested. The results show that the slump of a recycled concrete mixture is 120 mm when the content of recycled aggregate is less than 30%, and the mechanical strength satisfies the requirement of the high fluidity ordinary concrete. As the ratio of sand increases, the fluidity of the recycled concrete also improved. The best sand ratio is not consistent with the strength index. When the sand ratio is between 0.37 and 0.40 and the content of recycled aggregate is not more than 40%, the concrete of C60 can be prepared. Both the fluidity and the strength index can meet the design requirements and can be used in the practical engineering.

## 1. Introduction

There are many kinds of construction wastes which have caused serious different social and environmental problems. How to make full use of construction wastes is an important problem for the government and society. Hence, it is necessary to study the recycled concrete. The surface of the recycled aggregate is attached to a large amount of aging cement slurry, and this will increase the numbers of the pores and cracks, resulting in lower workability and lower mechanical properties of recycled concrete compared with natural concrete [1,2,3,4].

The scholars have used the physical and chemical methods to strengthen the surface of the recycled aggregates to make the performance of the recycled concrete reach or exceed the ordinary concrete. Al-Bayati et al. [5] combines a 350 °C heat treatment method and mechanical treatment method to improve the physical performance. The results show that the water absorption rate reduces more than 27% through the combined methods. Shi et al. [6] greatly increases the strength of the recycled concrete by using volcanic mortar. Zhong et al. [7] used a water glass solution, silane slurry and polyvinyl alcohol solution for single and composite impregnation of recycled coarse aggregates, and studied the effects of different modifiers and impregnation methods on the basic properties of recycled aggregates, and used SEM to observe the microstructure of the treated specimens, and the results showed that the composite impregnated treated recycled aggregates had better performance. Using the new recycled concrete aggregate disposed of by carbonation of 100%, Xuan et al. [8] obtain that the strength of the recycled concrete will not decrease as usual when the replacement ratio of recycled aggregates is 60%. These methods have high economic costs and complicated operations, which are not conducive to engineering promotion; and scholars think of the advantages of reinforced concrete research. Mastali et al. [9] improved the mechanical properties and impact resistance of recycled concrete based on increasing the volume fraction and length of carbon fiber. Chen et al. [10] demonstrated that the rational volume content of steel fiber is helpful to improve the ductility and crack performance of recycled concrete at high temperatures.

Both the fine and coarse aggregates can be replaced by the recycled aggregates in the practical engineering [11,12,13,14]. Abera [15] evaluated the overall benefits of recycled aggregate materials from different sources of construction and demolition waste, and in order to determine the optimal percentage of recycled aggregate, tests were conducted by varying the substitution rates of recycled coarse and fine aggregate materials. The results showed that there is a direct relationship between the performance of the concrete material and the recycled aggregate substitution rate. Jalilifar et al. [16] analyzed the microstructure of fully recycled coarse aggregate concrete using different types of mineral admixtures using SEM and showed that the control group using 10% silica significantly reduced the interfacial transition zone of porosity and compression by bridging hydration products, the volcanic ash reaction of fly ash and the low reactivity of natural zeolite powder made the concrete containing these two. The volcanic ash reaction of fly ash and the low reactivity of natural zeolite powder make the recycled concrete containing these two mineral admixtures have relatively porous and uncompressed microstructure and there are more discontinuities in the interfacial transition zone of this type of recycled concrete. To promote the application of recycled concrete in actual engineering and decrease the cost of high-quality recycled concrete materials, many improvement methods have been proposed by home and abroad scholars. Wang et al. [17] considered water absorption, water binder ratio and recycled aggregate sand ratio to prepare high-performance recycled concrete. Yang and Lee [18] use equivalent mortar volume method to prepare recycled concrete. Najim et al. [19,20,21] found that recycled concrete also can be used in the structures and the load bearing capacity performance of the members is studied. Muzaffer et al. [22] demonstrated that the addition of granulated blast furnace slag in recycled concrete, the mechanical strength of the concrete will be highly improved when the water-cement ratio is 0.5. Du et al. [23] investigated the dynamic mechanical properties of recycled and natural concrete by impact tests and numerical simulations, and the results showed that the dynamic mechanical properties of recycled concrete are good and the numerical model is feasible for qualitative analysis of SHPB impact tests of RC. Joyklad et al. [24] investigated the effect of using crushed brick waste as a substitute for natural coarse aggregate in concrete, comparing properties such as compressive strength, tensile splitting strength, flexural strength, modulus of elasticity, and stress-strain behavior of natural and recycled aggregate concrete. The results show that the greater the replacement of recycled aggregate, the lower the strength, but the ductility is partially increased and the overall performance is satisfactory.

The recycled aggregate also can be used in self-compacting concrete. There are many research achievements [25,26,27]. Martínez-García et al. [28] examined the effect of recycled coarse aggregate on the physical and mechanical properties of self-compacting concrete by comparing concrete with 20%, 50% and 100% recycled coarse aggregate replacement with plain concrete. The results showed that the recycled coarse aggregate self-compacting concrete showed no significant difference from the normal self-compacting concrete in terms of flexural strength, the tensile strength was reduced, the compressive strength varied with the substitution rate of recycled coarse aggregate and showed a reduction in strength when the natural aggregate was completely replaced. Singh N. et al. [29] studied the carbonation and resistance of coarse recycled concrete aggregate self-compacting concrete. The recycled concrete aggregate and recycled asphalt pavement are used in the production of self-consolidating concrete with varying percentage replacements of natural coarse aggregate. A total of 16 concrete mixtures were prepared and tested by Khodair [30].

High fluidity recycled concrete has a wide range of application, saves the use of natural aggregates, and is low-carbon and environmentally friendly. On the one hand, it can make secondary use of construction waste and reduce the use of natural aggregates, on the other hand, high fluidity can effectively solve the problem of difficult vibrating during concrete placement, which ensures the construction quality and provides a guarantee for the safety of buildings. However, there is very little research on high fluidity recycled concrete, which has a broad use prospect. For this reason, this paper prepares high fluidity recycled concrete with different matching ratios based on the matching ratio of high fluidity concrete with C60 natural aggregate, and conducts experimental research and theoretical analysis on the workability and mechanical properties of high fluidity recycled concrete to provide reference for the experimental research and engineering application of high fluidity recycled concrete.

## 2. Experimental Details

### 2.1. Materials

The apparent density of the river sand is 2596 kg/m^3^ and the fineness modulus is 2.48. The water reduction rate of the water reducing agent is 30%. Mixing water comes from ordinary tap water. Natural aggregate is gravel and the recycled aggregate is screened by artificial crushing. According to the “standard for technical requirements and test method of sand and crushed stone (or gravel) for ordinary concrete” (JGJ 52-2006) [31], the physical property of the natural and recycled aggregate is conducted and shown in Table 1. The Portland cement with P.O 42.5 grade is used in the mixture. According to “Common Portland cement” (GB 175-2007) [32], the physical property of the cement is shown in Table 2.

### 2.2. Experimental Method

The reference concrete is high fluidity natural concrete with a grade of C60. The slump is 200 mm. The admixture of the water reducing agent is 0.45%. The sand ratio is 0.38 and the mix proportion is C:W:S:G = 1:0.32:1.45:2.37. Based on the reference concrete, the workability and mechanical property of the high fluidity recycled concrete with different recycled aggregate content and sand ratio are carried out.

Firstly, the recycled aggregate content (20%, 30%, 40%, 50%, 60%) is changed to get the appropriate amount of recycled aggregate. Then the sand ratio (0.37, 0.38, 0.39, 0.40) is changed to prepare different high fluidity recycled concrete. The mix proportion is shown in Table 3.

The workability of the recycled concrete is conducted by a slump as shown in Figure 1. The specimens of the mechanical property test are non-standard block. The dimensions of the compression strength and tensile strength specimens are 100 mm× 100 mm× 100 mm. The dimensions of flexural strength specimens are 100 mm× 100 mm× 400 mm. The mechanical properties test uses non-standard test blocks, and the reduction factors of compressive strength, tensile strength, and flexural strength are 0.95, 0.95 and 0.85, respectively. The 3 d, 7 d, and 28 d strength tests are carried out according to the “Standard for test methods of concrete physical and mechanical properties” (GB/T 50081-2019) [33].

## 3. Results

### 3.1. Performance of Recycled Concrete with Different Recycled Aggregate Content

#### 3.1.1. Workability

The slump of the recycled concrete with different recycled aggregate content is shown in Figure 2.

It can be shown from Figure 2:(1)With the increase in recycled aggregate content, the slump of recycled concrete shows a decreasing trend. When the amount of recycled aggregate increases from 0 to 60%, the state of the mixture comes from a high fluidity state to a dry plastic state. If the recycled aggregate content is more than 60%, the high fluidity of the recycled concrete is very poor and cannot be used. This is the same as the results of Abera [15], where the substitution rate of recycled aggregates had a significant effect on mobility.(2)Due to the production technology, old mortar inevitably adheres to the surface of recycled aggregates [34]. These adhered mortars are an important reason for the reduced performance of recycled aggregates, which can lead to increased water demand and insufficient hydration of the cementing materials. The rough surface of the recycled aggregates leads to an increase in the friction of fresh concrete and a corresponding decrease in fluidity. With the increase in recycled aggregate replacement rate, this effect is more significant.(3)When the recycled aggregate content is not more than 30%, the slump of recycled concrete is 120 mm. Meanwhile, the slump satisfies the requirements of pumping concrete slump. When the amount of recycled aggregate is more than 30%, according to the test results, the liquidity is reduced significantly and is not suitable for use in engineering.

#### 3.1.2. Mechanical Property

The recycled aggregate suffers secondary damage, and it carries more pores and cracks, resulting in a lower strength. The unhydrated cement slurry on the surface of the recycled aggregate enhances the interface strength between the slurry and the aggregate. The damage models of compression, tension and flexion resistance are shown in Figure 3.

It can be observed from Figure 3 that there are more red sintered bricks on the damage surface of the compression strength test. Under the action of the load, the specimen first produces internal damage, and then gradually expands to the outer edge as the load increases, and gradually small cracks appear on the surface of the specimen, accompanied by the phenomenon of shedding the outer surface, the aggregate is exposed, and the cracks continue to expand until the specimen loses its load-bearing capacity. The damage pattern of the tensile test is that many tiny cracks appear in the middle of the specimen as the load increases. These tiny cracks develop continuously and converge into a crack through the whole specimen, and the specimen fractures into two parts. The crack development pattern of the bending test is a penetration crack at the bottom center of the specimen under load, and the crack expands upward continuously, and the expansion direction is biased towards one end of the specimen. The crack develops rapidly at the later stage of loading, and finally produces an oblique penetration crack with a large depth, and the specimen is damaged and loses the bearing capacity.

The measured results of the mechanical properties of recycled concrete with different recycled aggregate content are shown in Table 4.

It can be observed from Table 4:(1)With the increase in recycled aggregate content, the compressive, tensile and flexural strength of recycled concrete shows a downward trend. Compared with the natural concrete, the maximum reduction rates of 28 d compressive, tensile and flexural strength of RC-60 recycled concrete are 18.9%, 17.3%, and 52.1%, respectively. The results of the literature [28] showed that the recycled coarse aggregate self-compacting concrete did not show significant differences from ordinary self-compacting concrete in terms of flexural strength, the tensile strength was reduced, and the compressive strength varied with the replacement rate of recycled coarse aggregate. Analysis of the reasons leading to the different results concluded that the types of recycled aggregates used and the way they were made were different, and that the recycled aggregates were crushed and reused by the waste concrete, producing many cracks, resulting in the mechanical strength of the recycled concrete decreasing with the increase in recycled aggregates, thus showing different performance changes.(2)From the analysis of the compressive strength of the increase in age, the compressive strength of recycled concrete with different recycled aggregate content increases. Compared with its own 28 d compressive strength, the more recycled aggregate is added, the faster the early strength develops. There are many pores in the recycled aggregate, which provides sufficient space for early hydration. Meanwhile, the cement slurry is hydrated on the surface of the recycled aggregate. Therefore, the hydration speed is faster and the early strength development is higher.(3)When the replacement rate of recycled aggregates is less than 30%, the tensile strength of recycled concrete is not lower than that of ordinary concrete. This is similar to the law obtained by Zheng et al. [35], where a 30% recycled aggregate replacement rate is more suitable for preparing high-fluidity recycled concrete.

Figure 4 shows the measured results of the tension and compression ratio of recycled concrete with different recycled aggregate content. Figure 5 shows the measured results of bend, and the compression ratio of recycled concrete with different recycled aggregate content.

It can be observed from Figure 4 and Figure 5:(1)When the amount of recycled aggregate is not more than 40%, the tension and compression ratio of recycled concrete increases with the increase in the recycled aggregate concrete. When the amount of recycled aggregate is more than 40%, the tension and compression ratio of recycled concrete shows an inconsistent change. The reasons are as follows. Firstly, The presence of the recycled aggregate makes the compressive performance of recycled concrete is less than that of natural concrete. Secondly, the interfacial strength between the slurry and the bone of the recycled aggregate is higher than natural concrete. In the case of tensile failure, the recycled concrete is destroyed by the aggregate itself and the tensile strength is improved. Therefore, the tension and compression ratio of the recycled concrete increases.(2)With the increase in recycled aggregate content, the bend and the compression ratio of recycled concrete shows a gradual decline trend. Since the recycled aggregate has many pores and inside cracks, the stress concentration is likely to occur at the pores and cracks when the recycled concrete is under bending stress. The flexural strength of the recycled concrete decreases with the increase in the recycled aggregate content. Therefore, the bend and the compression ratio of recycled concrete is lower than that of natural concrete.

### 3.2. Performance of Recycled Concrete with Different Sand Ratio

#### 3.2.1. Workability

The measured recycled concrete slump of 0, 30%, 40% and 50% recycled aggregate content with the different sand ratio is shown in Figure 6.

It can be observed from Figure 6:(1)With the increase in sand ratio, the high fluidity of recycled concrete mixture of NC-0, RC-30, RC-40, and RC-50 is highly improved, but the amplitude is not significant. With the increase in sand ratio, the filling effect of sand and the ball effect becomes more and more obvious, and the high fluidity of recycled concrete increases. This is relatively similar with the results of the literature [36], but the sand rate used in this paper is relatively small, so the conclusion obtained is that the flow rate increases continuously with increasing sand rate.(2)The maximum increase value of the slump of NC-0, RC-30, RC-40, and RC-50 recycled concrete is 15 mm, 25 mm, 15 mm and 30 mm, respectively, with different sand ratios. Since the surface of the recycled aggregate is rough, and there are many pores and cracks, and the water absorption of the recycled aggregate is high, the workability of the recycled concrete will not be improved with different amounts of the sand.

#### 3.2.2. Mechanical Property

The 3 d, 7 d, and 28 d measured the compressive strength of the NC-0, RC-30, RC-40 and RC-50 recycled concrete with the different sand ratio is shown in Figure 7.

It can be seen from Figure 7:(1)Compared with the 28d compressive strength, the 3 d compressive strength of NC-0, RC-30, RC-40, and RC-50 specimens can reach 85%, and the 7 d compressive strength of them can reach 95%. The early strength of concrete is high, mainly due to the progress of current cement technology. The cement particle is much finer and more uniform, and the early hydration is faster and early compressive strength is high.(2)When the amount of recycled aggregate is not more than 30%, and the sand ratio is 0.40, the compressive strength of recycled concrete is close to the natural concrete. The 28 d compressive strength of recycled concrete with different recycled aggregate content is different with different sand ratio. The optimum sand ratio of NC-0 and RC-40 is 0.37, and the optimum sand ratio of RC-30 is 0.40. The optimum sand ratio of RC-50 is 0.38.

The measured tensile strength and the flexural strength of the NC-0, RC-30, RC-40, and RC-50 recycled concrete with different sand ratios are shown in Figure 7.

It can be seen from Figure 8 and Figure 9:(1)The tensile strength of RC-30 and RC-40 recycled concrete is closed to NC-0 concrete with a rational sand ratio exceed the tensile strength of NC-0 concrete. Compared with the flexural strength of the NC-0, the maximum flexural strength of RC-30, RC-40, and RC-50 concrete decreases by 24.5%, 28.5% and 34.0%, respectively.(2)The maximum compressive strength, tensile strength and the flexural strength of the RC-30 are 60.2 MPa, 3.97 MPa, and 5.42 MPa, respectively, when the recycled aggregate content does not exceed 30%. The results satisfy the requirements of the “Specification for mix proportion design of ordinary concrete” (JGJ 55-2011) [37].(3)The change in sand ratio does not improve the mechanical properties of with different recycled aggregate content. The rational sand ratio can improve the compactness of the recycled concrete. This is in agreement with the results of literature [36], where proper sand ratio helps to improve the compactness of concrete. However, the strength of recycled concrete is lower than that of ordinary concrete because recycled aggregate is inferior to natural aggregate in material properties. This is also reflected in document [38], which is an unavoidable problem when using recycled aggregate.

Figure 10 shows the results of the tension and compression ratio of NC-0, RC-30, RC-40 and RC-50 recycled concrete with different recycled aggregate content. Figure 11 shows the measured results of bend and the compression ratio of NC-0, RC-30, RC-40 and RC-50 recycled concrete with different recycled aggregate content.

It can be seen from Figure 10 and Figure 11:(1)The tension and compression ratio and the bend and compression ratio of recycled concrete with different sand ratios show inconsistent changes. The tension and compression ratio is about 6.0%, the bend and compression ratio is between 8.0% and 10.0% with different recycled aggregate content. The tension and compression ratio of natural concrete is between 10.0% and 11.2%.(2)From the comparison of the tensile and compressive ratios of different recycled concretes with the different sand ratio, the incorporation of recycled aggregates can improve the toughness of recycled concrete. The strength of the recycled aggregate is low by its own characteristic which is not conducive to the toughness of recycled concrete.

In this paper, the effects of recycled aggregate replacement rate and sand rate on the workability and mechanical properties of high fluidity recycled concrete were investigated, and certain results were achieved, but there is still much space for research. In the subsequent research, the state of the transition zone at the interface of recycled aggregate and cement paste, the microscopic change morphology of recycled aggregate under load, mixing admixtures and fibers, etc., should be analyzed according to SEM.

## 4. Conclusions

(1)Compared with high fluidity natural concrete, there is an obvious change in the workability and mechanical strength of recycled concrete when the amount of recycled aggregate is more than 60%. According to the flexural performance results, the brittleness characteristic of the recycled concrete is effectively improved when the recycled aggregate content is between 30% and 50%.(2)The increase in sand ratio can increase the high fluidity of the recycled concrete maximum up to 30 mm. The slump of a recycled concrete mixture is 120 mm when the content of recycled aggregate is less than 30%, and the mechanical strength satisfies the requirement of the high fluidity ordinary concrete.(3)Along with the increase in sand ratio, the fluidity of the recycled concrete also improved. The best sand ratio is not consistent with the strength index. When the sand ratio is between 0.37 and 0.40 and the content of recycled aggregate is not more than 40%, the concrete of C60 can be prepared. Both the fluidity and the strength index can meet the design requirements and can be used in the practical engineering.

## Figures and Tables

**Figure 1 materials-15-06104-f001:**
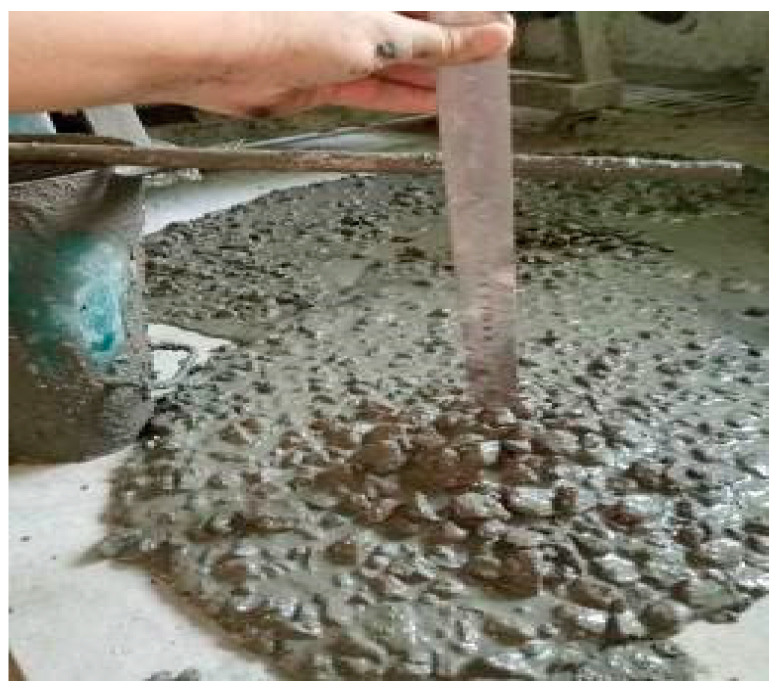
Slump test of the concrete.

**Figure 2 materials-15-06104-f002:**
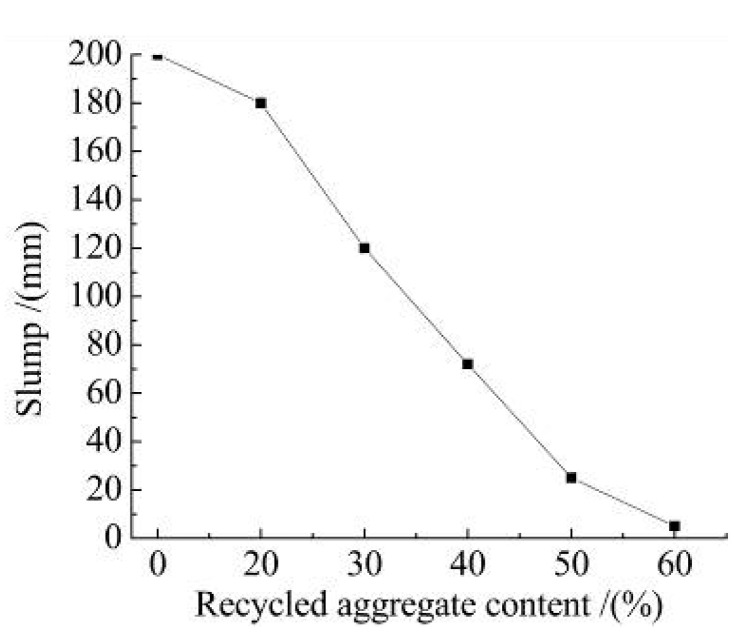
The slump of recycled concrete with different recycled aggregate content.

**Figure 3 materials-15-06104-f003:**
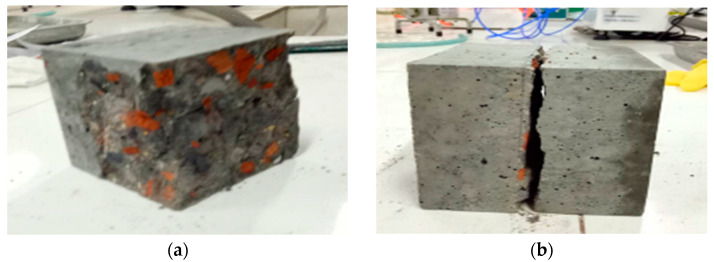
(**a**) Compression strength (**b**) Tensile strength (**c**) Flexural strength.

**Figure 4 materials-15-06104-f004:**
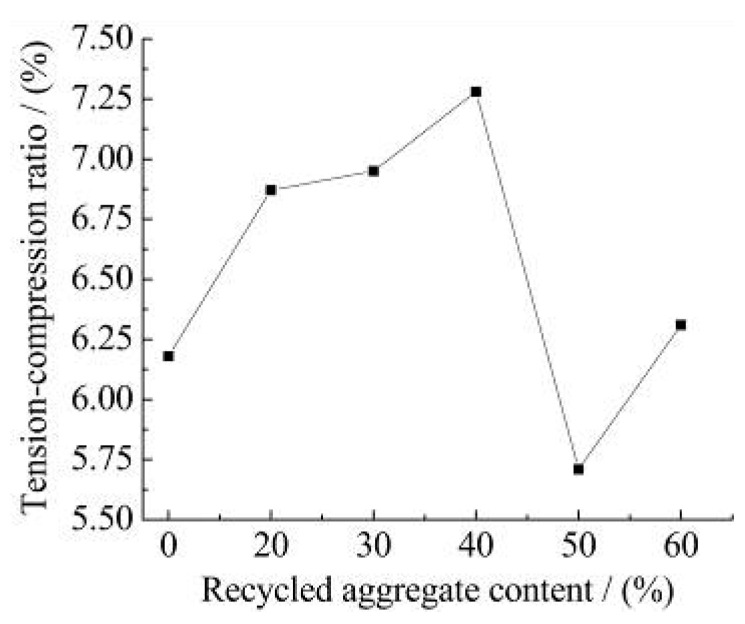
Tension and compression ratio.

**Figure 5 materials-15-06104-f005:**
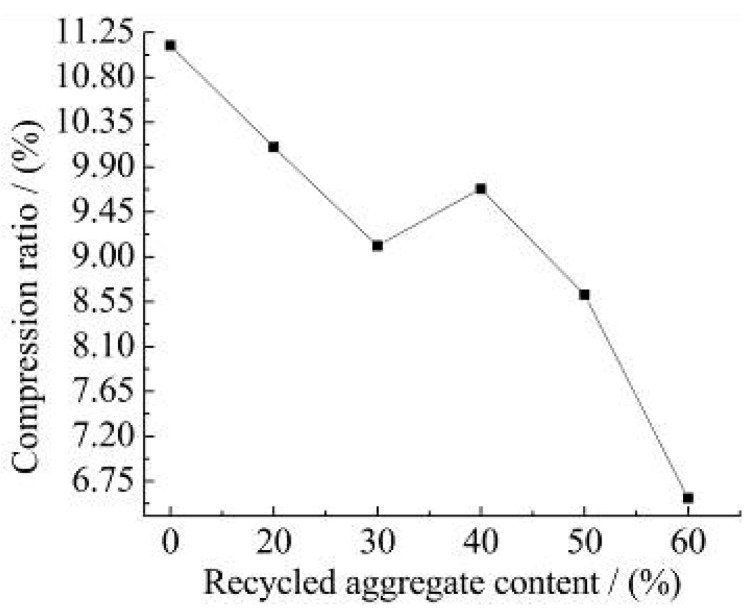
Bend and the compression ratio.

**Figure 6 materials-15-06104-f006:**
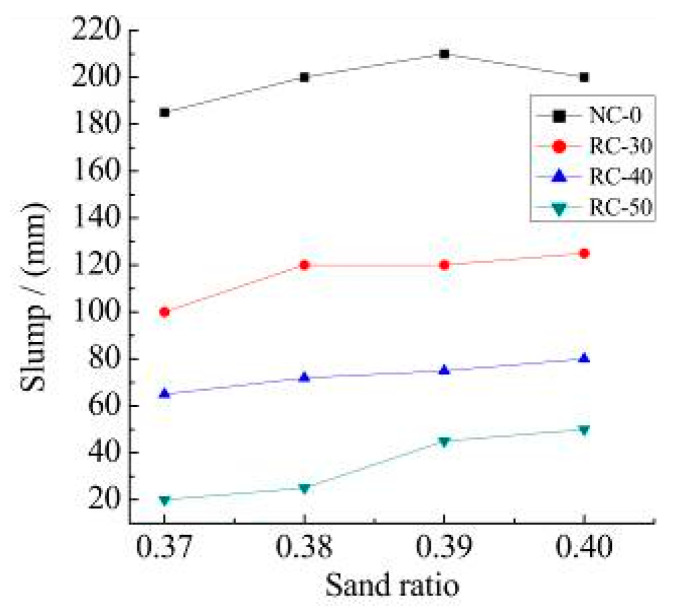
Measured recycled concrete slump with the different sand ratio.

**Figure 7 materials-15-06104-f007:**
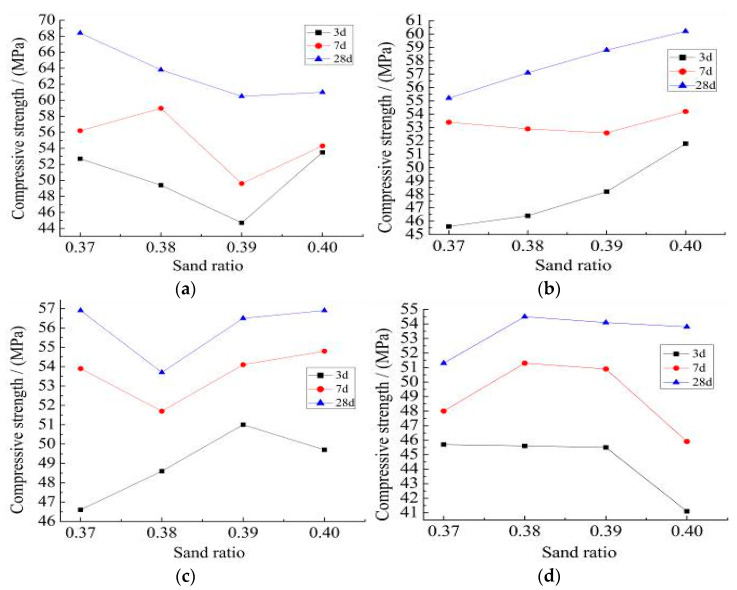
(**a**) NC-0, (**b**) RC-30, (**c**) RC-40, (**d**) RC-50.

**Figure 8 materials-15-06104-f008:**
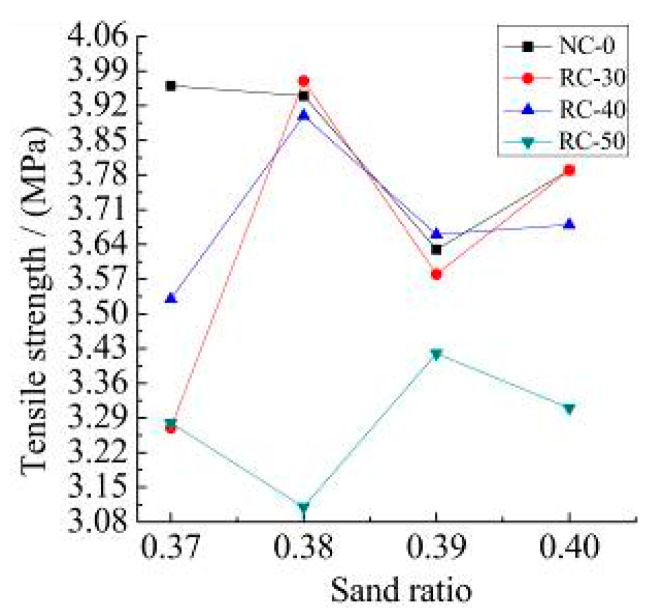
Measured tensile strength.

**Figure 9 materials-15-06104-f009:**
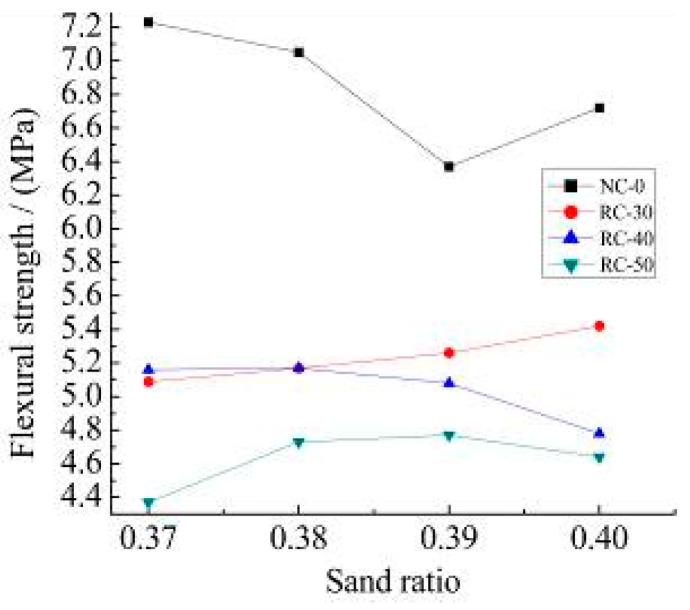
Measured flexural strength.

**Figure 10 materials-15-06104-f010:**
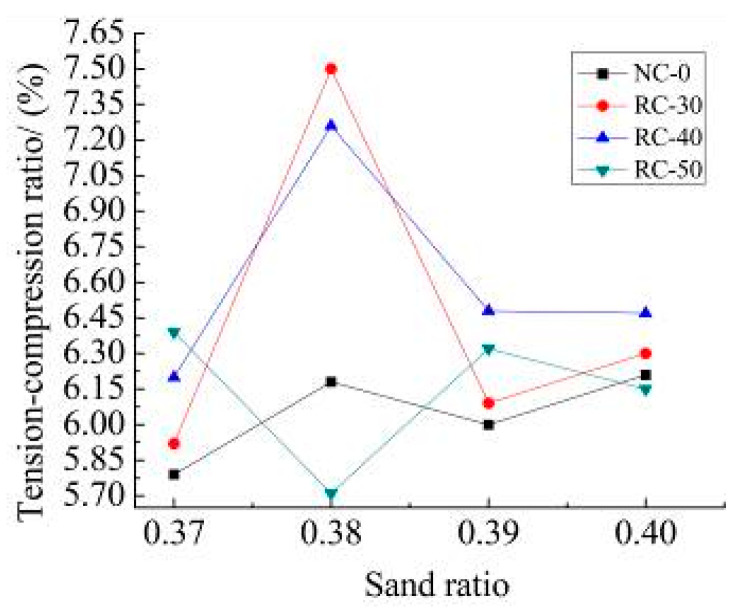
Tension and compression ratio.

**Figure 11 materials-15-06104-f011:**
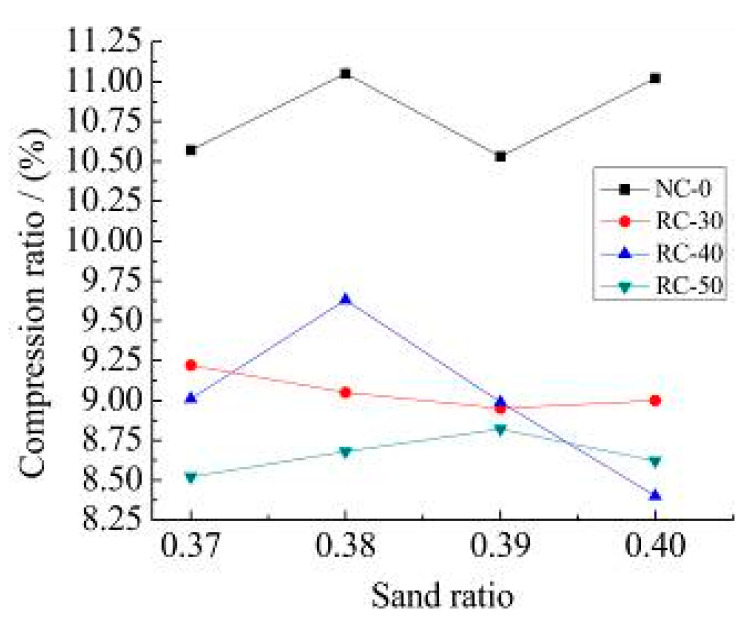
Bend and compression ratio.

**Table 1 materials-15-06104-t001:** Physical property of the coarse aggregates.

Material	Apparent Density (g/cm^3^)	Bulk Density (g/cm^3^)	Moisture Content (%)	Particle Grading
Natural aggregate	2.73	1.6	0.06	5~25
Recycled aggregate	2.55	1.46	7.38	5~25

**Table 2 materials-15-06104-t002:** Physical property index of the cement.

Density (g/cm^3^)	Specific Surface Area (m^2^/kg)	Mortar Fluidity (mm)	Water Requirement of Normal Consistency (%)	Fineness (%) (>80 μm)	Setting Time (min)	
					Primary Coagulation	Final Condensation
3.02	328	192	29.8	0.31	126	208

**Table 3 materials-15-06104-t003:** Mix proportion of the concrete.

Specimens	Recycled Aggregate Content (%)	Sand Ratio	Natural Aggregate (kg)	Water-Cement Ratio	Cement (kg)	Water Reducing Agent Volume Ratio (%)
NC-0	0	0.37	1120	0.32	465	0.45
NC-0	0	0.38	1100	0.32	465	0.45
NC-0	0	0.39	1080	0.32	465	0.45
NC-0	0	0.40	1060	0.32	465	0.45
RC-20	20	0.38	880	0.32	465	0.45
RC-30	30	0.37	785	0.32	465	0.45
RC-30	30	0.38	770	0.32	465	0.45
RC-30	30	0.39	755	0.32	465	0.45
RC-30	30	0.40	740	0.32	465	0.45
RC-40	40	0.37	670	0.32	465	0.45
RC-40	40	0.38	660	0.32	465	0.45
RC-40	40	0.39	650	0.32	465	0.45
RC-40	40	0.40	640	0.32	465	0.45
RC-50	50	0.37	560	0.32	465	0.45
RC-50	50	0.38	550	0.32	465	0.45
RC-50	50	0.39	540	0.32	465	0.45
RC-50	50	0.40	530	0.32	465	0.45
RC-60	60	0.38	440	0.32	465	0.45

**Table 4 materials-15-06104-t004:** Measured results of the mechanical property of recycled concrete with different recycled aggregate content.

Specimens	Compressive Strength/MPa			Tensile Strength/MPa	Flexural Strength/MPa
	3 d	7 d	28 d	28 d	28 d
NC-0	49.4	59.0	63.8	3.94	7.1
RC-20	50.1	54.3	58.4	4.01	5.9
RC-30	46.4	52.9	57.1	3.97	5.2
RC-40	48.6	51.7	53.6	3.90	5.2
RC-50	45.6	51.3	54.5	3.11	4.7
RC-60	40.5	41.4	51.7	3.26	3.4

## Data Availability

Not applicable.

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
