# Peer review of "Experimental Study and Analysis on Workability and Mechanical Performance of High Fluidity Recycled Concrete"

_materials, 2022, doi:10.3390/ma15176104_

Round 1

Reviewer 1 Report

The paper has no scientific grounds. It is merely a report of standard laboratory tests on small concrete specimens with varying amounts of recycled material. It lacks originality. The manuscript cannot be accepted in its current form.

Full stress-strain curves during compression and load-deflection curves during bending must be included in the article (or load-displacement curves during splitting). The influence of varying recycled aggregate content on concrete brittleness (an essential concrete attribute) must be demonstrated.

Concrete specimen failure mechanisms (crack patterns) (and their progression) must be sketched in detail.

The importance of the experimental results for engineering practice must be thoroughly explained.

The experimental findings must be compared to previous tests on concrete with various waste recycled aggregate from the literature.

Most of conclusions are trivial.

Author Response

20-Aug-2022

Dear Editor and reviewers,

Thank you for your letter and professional comments on our manuscript entitled “Experimental Study and Analysis on Workability and Mechanical Performance of High Flowing Recycled Concrete” (materials-1864085). The comments are very helpful for revising and improving our manuscript. We have studied comments carefully and have revised the manuscript. All revised parts have been marked up using the “Track Changes” function.

The main revisions in the paper and the responses to the reviewers’ comments are shown as follows.

We hope the revisions meet all the requirements. We are looking forward to your positive reply.

Sincerely,

Cun Hui

Zhongyuan University of Technology

______________________________________________________________

Reviewer #1:

The paper has no scientific grounds. It is merely a report of standard laboratory tests on small concrete specimens with varying amounts of recycled material. It lacks originality. The manuscript cannot be accepted in its current form.

Reply:

Thanks for your professional comments. We carefully studied the comments and revised the manuscript.

The main objective of this paper is to prepare high flowing recycled concrete with excellent workability by using recycled aggregates instead of natural aggregates on the basis of high flowing plain concrete by varying the replacement rate (20%, 30%, 40%, 50%, 60%) and sand rate (0.37, 0.38, 0.39, 0.40) of recycled aggregates.

Recycled aggregate is made of construction waste crushing, which can be used again after the demolition of old buildings. It is an effective way to treat and resource construction waste, which is in line with the current environmental protection concept of harmonious coexistence between human beings and nature, a reflection of the development concept of innovation, coordination, green, openness and sharing in the construction industry, and a necessary path for green and low-carbon development.

Therefore, we believe that the reasonable use of construction waste to prepare recycled aggregates and replace natural aggregates can effectively alleviate the current worldwide problem of construction waste disposal and realize the secondary use of construction waste, which has certain application value.

  1. Full stress-strain curves during compression and load-deflection curves during bending must be included in the article (or load-displacement curves during splitting). The influence of varying recycled aggregate content on concrete brittleness (an essential concrete attribute) must be demonstrated.

Reply:

The load-deflection curves during bending have been explained in the other manuscripts. In order to avoid the repetition, the curves were not put in this manuscript.

  1. Concrete specimen failure mechanisms (crack patterns) (and their progression) must be sketched in detail.

Reply:

We have added the failure modes in subsection 3.1.2.

  1. The importance of the experimental results for engineering practice must be thoroughly explained.

Reply:

Comparing and analyzing our experimental results, we obtained the suitable recycled aggregate replacement rate and sand rate within the test range to prepare high flowing recycled concrete, which provides a reference for the engineering application of high flowing recycled concrete.

The use of recycled aggregates is an effective way to treat and resource construction waste, which is in line with the current environmental protection concept of harmonious coexistence between man and nature, a reflection of the development concept of innovation, coordination, green, openness and sharing in the construction industry, and a necessary path to green and low-carbon development.

Concrete plays an important role in the construction industry and almost all buildings require concrete to be poured. However, there are certain difficulties in the process of concrete placement that need to be solved, for example, some parts of the reinforcement are dense and the concrete is difficult to vibrate, which can lead to construction quality problems. High flowing recycled concrete can appropriately solve this problem. Concrete with good fluidity will have lower requirements for vibrating, which indirectly improves the quality of construction.

Therefore, the experiments conducted in this paper prepared high flowing recycled concrete and obtained suitable recycled aggregate replacement rate and sand rate, which can provide reference for the engineering application of recycled concrete and effectively solve the problems of construction waste disposal and concrete placement.

  1. The experimental findings must be compared to previous tests on concrete with various waste recycled aggregate from the literature.

Reply:

We have compared the results of previous studies in the results.

  1. Most of conclusions are trivial.

Reply:

The authors have regrouped the conclusions.

Reviewer 2 Report

Experimental Study and Analysis on Workability and Mechanical Performance of High Flowing Recycled Concrete

This study presents workability and mechanical performance of the recycled concrete.

Objectives and aims are missing in the abstract.

Some more latest studies are required in the introduction section to further highlight the importance of this study.

Effect of Composite Impregnation on Properties of Recycled Coarse Aggregate and Recycled Aggregate Concrete. Buildings, 12(7), 1035.

Investigation on Dynamic Mechanical Properties of Recycled Concrete Aggregate under Split-Hopkinson Pressure Bar Impact Test. Buildings, 12(7), 1055.

EFFECT OF FIRED CLAY BRICK AGGREGATES ON MECHANICAL PROPERTIES OF CONCRETE. Suranaree Journal of Science & Technology, 25(4).

Section 2.1 How material properties were collected.

Section 2.2, how percentages were selected, there should be proper replacements following previous studies.

Figure 1, slump values seems to very high than as reproted in Figure 2.

Overall the quality of the figures is very low.

Authors must summarized results in more systematic way with reference to the previous studies.

Also, Conclusions are too limited to proof the significant outcome of this study.

Author Response

20-Aug-2022

Dear Editor and reviewers,

Thank you for your letter and professional comments on our manuscript entitled “Experimental Study and Analysis on Workability and Mechanical Performance of High Flowing Recycled Concrete” (materials-1864085). The comments are very helpful for revising and improving our manuscript. We have studied comments carefully and have revised the manuscript. All revised parts have been marked up using the “Track Changes” function.

The main revisions in the paper and the responses to the reviewers’ comments are shown as follows.

We hope the revisions meet all the requirements. We are looking forward to your positive reply.

Sincerely,

Cun Hui

Zhongyuan University of Technology

______________________________________________________________

Reviewer #2:

  1. Objectives and aims are missing in the abstract.

Reply:

The authors have revised the abstract and relevant content has been added.

  1. Some more latest studies are required in the introduction section to further highlight the importance of this study.

Reply:

The authors have added recent research advances in recycled concrete based on the recommendations.

  1. Section 2.1 How material properties were collected.

Reply:

The testing of material properties was carried out according to the Chinese specifications "Standard for technical requirements and test method of sand and crushed stone (or gravel) for ordinary concrete" (JGJ 52-2006) and "Common portland cement" (GB 175-2007). Detailed methods for testing material properties are provided in the specifications.

  1. Section 2.2, how percentages were selected, there should be proper replacements following previous studies.

Reply:

The C60 grade high flowability concrete used in this paper is the result of our group's research, and no substitution test of recycled aggregates was conducted before that. The authors prepared high-flowing recycled concrete with different recycled aggregate replacement rates in a test mix one by one, and varied the sand rate in parallel tests to prepare high-flowing recycled concrete with excellent performance. After comparing and analyzing the effects of recycled aggregate replacement rate and sand rate on flowability, the best mix ratio within the test range was obtained.

  1. Figure 1, slump values seems to very high than as reproted in Figure 2.

Reply:

The slump values of the Figure 1 is about 200mm, it seems to be higher than reported because of photographing angle.

  1. Overall the quality of the figures is very low.

Reply:

Most of the figures have been revised.

  1. Authors must summarized results in more systematic way with reference to the previous studies.

Reply:

The authors have supplemented the experimental results with relevant research results.

  1. Also, Conclusions are too limited to proof the significant outcome of this study.

Reply:

The authors have revised the conclusions.

Reviewer 3 Report

Manuscript ID: materials-1864085

Title: Experimental Study and Analysis on Workability and Mechanical Performance of High Flowing Recycled Concrete

Journal: materials

Comments to authors:

In the present study, the workability and mechanical performance of the recycled concrete are investigated to prepare high flowing recycled concrete by changing the amount of recycled aggregate (20%, 30%, 40%, 50% and 60%) and the sand ratio (0.37, 0.38, 0.39 and 0.40) based on the standard of high flowing ordinary concrete. A limited work with small novelty.

Recommendation: This manuscript can be accepted after a major revision. The authors are requested to address the following comments for the improvement of the manuscript:

The Abstract should be enriched with the brief details of the methodology.

What is the need for this work?

The authors should present more numeric results in the Abstract.

English proofreading should be done for grammar and typos such as “The sand ratio is 0.38 and the mix proportion is C: W: S: G=1: 0.32: 1.45: 2.37.” in this sentence, there should be a space between ‘G’ and ‘=’. Please correct kg/m3 also.

The novelty, scope, and significance of the present work should be highlighted in the last paragraph of the Introduction section.

There should be a space between numeric values and units.

How much samples were fabricated for each mix?

The authors are recommended to add more literature review on relevant works.

Is this work helpful for practical applications? Which practical applications?

The literature review should be improved by adding latest references and discussion.

SEM should be added to study the internal structure of such concrete composite.

Split-tensile strength property of manufactured concrete is also recommended to be added.

All Figures can be presented in a better way.

Results section should be defended using technical reasons and relevant references.

Conclusions look like a lab report. They should be refined and briefly presented. Some numerical results should be added.

The authors should add the future recommendations based on the present study.

Author Response

20-Aug-2022

Dear Editor and reviewers,

Thank you for your letter and professional comments on our manuscript entitled “Experimental Study and Analysis on Workability and Mechanical Performance of High Flowing Recycled Concrete” (materials-1864085). The comments are very helpful for revising and improving our manuscript. We have studied comments carefully and have revised the manuscript. All revised parts have been marked up using the “Track Changes” function.

The main revisions in the paper and the responses to the reviewers’ comments are shown as follows.

We hope the revisions meet all the requirements. We are looking forward to your positive reply.

Sincerely,

Cun Hui

Zhongyuan University of Technology

______________________________________________________________

Reviewer #3:

In the present study, the workability and mechanical performance of the recycled concrete are investigated to prepare high flowing recycled concrete by changing the amount of recycled aggregate (20%, 30%, 40%, 50% and 60%) and the sand ratio (0.37, 0.38, 0.39 and 0.40) based on the standard of high flowing ordinary concrete. A limited work with small novelty.

Recommendation: This manuscript can be accepted after a major revision. The authors are requested to address the following comments for the improvement of the manuscript:

  1. The Abstract should be enriched with the brief details of the methodology.

Reply:

We have re-summarized the abstract.

  1. What is the need for this work?

Reply:

The use of recycled aggregates is an effective way to treat and resource construction waste, which is in line with the current environmental protection concept of harmonious coexistence between man and nature, a reflection of the development concept of innovation, coordination, green, openness and sharing in the construction industry, and a necessary path to green and low-carbon development.

Concrete plays an important role in the construction industry and almost all buildings require concrete to be poured. However, there are certain difficulties in the process of concrete placement that need to be solved, for example, some parts of the reinforcement are dense and the concrete is difficult to vibrate, which can lead to construction quality problems. High flowing recycled concrete can appropriately solve this problem. Concrete with good fluidity will have lower requirements for vibrating, which indirectly improves the quality of construction.

The high flowing recycled concrete studied in this paper can, on the one hand, make secondary use of construction waste and reduce the use of natural aggregates, and on the other hand, the high fluidity can effectively solve the problem of difficult vibrations during concrete placement, ensuring the quality of construction and providing a guarantee for the safety of buildings.

  1. The authors should present more numeric results in the Abstract.

Reply:

We have re-summarized the abstract.

  1. English proofreading should be done for grammar and typos such as “The sand ratio is 0.38 and the mix proportion is C: W: S: G=1: 0.32: 1.45: 2.37.” in this sentence, there should be a space between ‘G’ and ‘=’. Please correct kg/m3 also.

Reply:

We have proofread the whole text.

  1. The novelty, scope, and significance of the present work should be highlighted in the last paragraph of the Introduction section.

Reply:

We have revised the introduction.

  1. There should be a space between numeric values and units.

Reply:

Thank you for your suggestion, we have corrected this error.

  1. How much samples were fabricated for each mix?

Reply:

Three samples were fabricated for each mix.

  1. The authors are recommended to add more literature review on relevant works.

Reply:

The authors have added the latest research advances in recycled concrete.

  1. Is this work helpful for practical applications? Which practical applications?

Reply:

The high flowing recycled concrete studied in this paper can, on the one hand, make secondary use of construction waste and reduce the use of natural aggregates, and on the other hand, the high fluidity can effectively solve the problem of difficult vibrations during concrete placement, ensuring the quality of construction and providing a guarantee for the safety of buildings.

This research can provide a reference for the engineering application of high flowing recycled concrete, solving the problem of construction waste disposal and concrete vibrating. This is a direction of green development in the construction industry and has great application potential.

  1. The literature review should be improved by adding latest references and discussion.

Reply:

The authors have added the latest research advances in recycled concrete.

  1. SEM should be added to study the internal structure of such concrete composite.

Reply:

Thanks for your suggestion, and we will consider adding SEM to observe and analyze the internal structure of high flowing recycled concrete in subsequent tests.

  1. Split-tensile strength property of manufactured concrete is also recommended to be added.

Reply:

The splitting tensile strength properties of concrete have been explained in other manuscripts. To avoid repetition, there is not much explanation in this article.

  1. All Figures can be presented in a better way.

Reply:

Most of the figures have been revised.

  1. Results section should be defended using technical reasons and relevant references.

Reply:

The authors have supplemented the experimental results with relevant research results.

  1. Conclusions look like a lab report. They should be refined and briefly presented. Some numerical results should be added.

Reply:

The authors have regrouped the conclusions.

  1. The authors should add the future recommendations based on the present study.

Reply:

On the basis of this test, we have supplemented the follow-up research direction of high fluidity recycled concrete.

In this paper, the effects of recycled aggregate replacement rate and sand rate on the workability and mechanical properties of high flowing recycled concrete were investigated, and certain results were achieved, but there is still much space for research. In the subsequent research, the state of the transition zone at the interface of recycled aggregate and cement paste, the microscopic change morphology of recycled aggregate under load, etc. should be analyzed according to SEM.

Round 2

Reviewer 1 Report

I don't see the improvement of the manuscript. You didn't adopt  all of my comments. 

Reviewer 3 Report

The authors tried to address the comments. This work can be anyhow accepted now.